# Monkeypox Mysteries of the New Outbreak in Non-Endemic Areas

**DOI:** 10.3390/ijerph192214881

**Published:** 2022-11-12

**Authors:** Francisco Antunes, Ana Virgolino

**Affiliations:** 1Instituto de Saúde Ambiental, Faculdade de Medicina da Universidade de Lisboa, 1649-028 Lisboa, Portugal; 2Laboratório Associado TERRA, Faculdade de Medicina da Universidade de Lisboa, 1649-028 Lisboa, Portugal

**Keywords:** monkeypox virus, 2022 outbreak, men who have sex with men, smallpox vaccination, clades, emerging zoonosis

## Abstract

Monkeypox virus (MPXV) was declared by the World Health Organization (WHO) in mid-2022 to be a public health emergency of international concern, following its spread around the world after circulating in Western and Central Africa. This new outbreak is concentrated in men who have sex with men (MSM). Moreover, beyond the epidemiological change, compared with endemic countries, differences in clinical features and many other aspects have also been detected. These and other characteristics are unusual and still unclear. Based on the available data, the authors try to help to clarify some of the current major gaps in monkeypox knowledge to strengthen the outbreak response.

Human monkeypox virus (MPXV) belongs to the same poxviridae family and orthopoxvirus genus as variola virus (VARV), the causative agent of smallpox. The virus was first discovered at an animal facility in Copenhagen, Denmark, in 1958 [1]. Genome sequencing of MPVX confirmed it to be a distinct species of orthopoxvirus that is neither a direct ancestor nor a direct descendent of VARV [2]. The disease caused in humans by MPXV is similar to smallpox but much less severe. Human monkeypox was first diagnosed in a child in the Democratic Republic of Congo (DRC), in Central Africa, in 1970, while smallpox was later eradicated in 1980 [3]. Human monkeypox is a zoonotic disease endemic to areas of West and Central Africa, where a close interaction between humans and wild animals or reservoir(s) is common. The virus has two characterized genetic clades, the Western and the Central African clade [4]. In an average year, a few thousand human monkeypox cases occur in the endemic regions of West and Central Africa, and until 2022, cases outside these regions were limited and associated with the importation of infected rodents (2003) or travel to Africa (2018 and 2019) [5,6,7,8]. Based on the available data, this review aims to help understand the current major gaps in human understanding of the dynamics of viral transmission and epidemiological characteristics of this 2022 human monkeypox outbreak.

In May 2022, the multi-country outbreak of human monkeypox started to spread to Europe and North America, and in July 2022 it was determined to be a public health emergency of international concern (PHEIC) by the World Health Organization (WHO) [9]. Since it was discovered, and till 2022, monkeypox circulating in remote areas of several African countries was considered a rare human disease, despite the recognition that the number of cases had increased recently in two countries of West and Central Africa [10,11,12]. Human monkeypox reported in West and Central Africa features differences in transmissibility and disease severity, in West Africa being less human-to-human transmissible and less severe. One explanation for these relative differences would be the in vivo viral kinetics, with a prolonged and higher magnitude of viraemia in human monkeypox cases from Central Africa compared with those from West Africa. The genetic differences between the West African strain compared with the Central African strain could explain the observed differences [4].

Since May 2022, the number of cases identified outside Africa has already surpassed the total number detected in African countries, where the disease remained endemic in the last five decades after 1970, when the first case of human monkeypox was identified [13]. This current global outbreak, with thousands of cases, till now, fortunately, has led to only a few deaths or serious cases which require hospital admission [4,14]. The phylogenetic analysis of the genomic sequences of the MPXV strain, associated with this outbreak, shows that it belongs to the Western African clade [15].

There are several uncommon aspects of this 2022 human monkeypox outbreak. First, until very recently, it was unknown how much the strain causing this new outbreak is different from the clade circulating in West Africa [12,16], and whether the cases were linked to one another. It is worth noting that the MPXV variants were recently named, wherein the former Central Africa clade is Clade I, and the former Western African is Clade II [17]. Clade II consists of two subclasses, Clade IIa and Clade IIb, with the latter associated with the group of variants of this new outbreak [17]. Characterization of all MPXV ongoing outbreak genome sequences, confirming a linkage between the 2022 MPXV lineage with the clade IIa (former Western African clade), suggests that the outbreak had a single origin [15]. However, this origin will have diverged by polymorphisms and several mutations, making the virus more transmissible. These genetic findings suggest that the MPXV strain has been spreading from a single origin, following human-to-human chains, due to close physical contact with infected people, without substantial travel links to monkeypox-endemic areas of West Africa, amplified by large gatherings and increased international travel after a containment period due to the COVID-19 pandemic [14,15,18]. It would be useful to sequence the virus in wild animals to better understand the virus’ genetic evolution. However, this will be very hard, because the natural reservoir(s) of MPXV in endemic regions of Africa is unknown [19].

Second, as mentioned, the number of monkeypox cases detected in this 2022 human monkeypox outbreak surpassed those in monkeypox-endemic regions of West and Central Africa [13]. The reasons for this increase require delineation through a One Health (human, animal, and environmental) approach as a zoonotic infection arising from the constant remodelling of ecosystems and continued human encroachment into animal habitats [18]. In West and Central Africa, the risk of monkeypox human outbreaks has been growing every year since 1980, when smallpox vaccination ended, meaning transmission chains have had the potential to spread further every year since then [11,12]. The new variant (clade IIb), with several mutations than would be expected, may promote changes in the biological aspects of the virus, including increased transmissibility, and together with superspreader events, and travel abroad likely triggered the rapid worldwide dissemination [14,15]. On the other hand, using an individual-based mathematical modelling framework, the introduction of three human monkeypox cases in a country could cause 18 secondary cases, 30 could cause 118 secondary cases, and 300 cases could cause 402 secondary cases [20]. These scientific, environmental, and social reasons may explain the recent unexpected global dissemination of human monkeypox cases beyond Africa.

Third, almost all the human monkeypox case clusters included men aged 20–50 years old [14,21,22]. Cases of monkeypox in Africa were comparatively small before the 1980s, when smallpox vaccination was discontinued worldwide. The smallpox vaccination provides cross-immunity against another poxvirus, including the MPXV [23,24]. The end of the global smallpox vaccination campaign in 1977 has resulted in a susceptible population because most estimates suggest immunity from the smallpox vaccination lasts three to five years and decrease immunity thereafter, and beyond that time its ability to protect against human monkeypox also may decrease, and the human-to-human cycles of transmission can occur [25,26,27]. This may explain the approximately 20-fold increase in monkeypox transmission, between 2006 and 2007, in the DRC and the 21-fold lower risk of infection for those who were vaccinated against smallpox since the 1980s [11]. Also, it is meaningful that the re-emergence of the human monkeypox outbreak in Nigeria, starting in 2017, in patients aged 21–40 years, with a median age of 31 years, has been similar to that observed in the DRC [12]. It is worth highlighting that all human monkeypox patients were born after 1978, when the smallpox vaccination campaign, starting in 1958, was over [28]. This new pattern of the disease in these endemic countries of Africa is assumed to be a consequence of waving cross-protective immunity, following smallpox vaccination discontinuation [11,19]. In this recent outbreak, human monkeypox infections have been confirmed to be present in persons with a median age of 38 years, compared to what was observed in West and Central Africa, which might reflect the growth of the population with weakened or no immunity against MPXV infection [14]. It is estimated that over 70% of the world’s population is no longer protected against smallpox, and deductively therefore against human monkeypox [29]. Concerning the transmission in the MSM community, whether monkeypox is sexually transmitted requires further studies. Sexual activity largely in MSM, in this ongoing outbreak, is the suspected route of transmission, supported by the unusual findings of lesions in genital, anal, and oral locations. A prolonged seminal MPXV DNA shedding and a live and replication-competent virus isolated from the semen sample might suggest a possible reservoir of the virus [30]. These findings support the likely transmission of MMPXV during sexual activity, though further studies are needed to improve the knowledge of the potential role of semen fluid transmission in the spreading of monkeypox infection. Overall, obtaining a clearer idea of the outbreaks and the risk factors for infection will involve rigorous contact tracing. The explanation for this pattern of the 2022 human monkeypox outbreak, could be the attendance at large gatherings linked to sex-onsite activities and international travel, probably starting when the virus was introduced into an MSM community, thus spreading easily in a non-immune population, with transmission amplified through sexual networks [31]. In the past, several epidemics of viral infections have been previously described at mass gathering events [31].

To conclude, the 2022 outbreak of human monkeypox going globally represents the most recent emerging zoonotic disease, where the clade IIb, primarily related to this outbreak, clusters with clade IIa (West African), and is closely related to the recently exported cases from Nigeria to the United Kingdom, Israel, and Singapore in 2018–2019, suggesting connections to a single case. The lineage that caused this 2022 outbreak differs from the one in West Africa. However, it is unclear whether this virus has mutated, whether that allows MPXV to become more transmissible and adapted to human-to-human transmission, and whether the rapid spread of the virus in various communities worldwide are linked to one another. A better understanding of the viral transmission chains and how the virus evolves is needed to contain the outbreak. However, this will be incredibly difficult because the monkeypox genome is a huge relative of that of many other viruses, and the natural animal reservoir(s) have yet to be discovered. How the ongoing outbreak started remains unclear, however, given the phylogenomic characterization of the virus and the MPVX historical epidemiology (rarity of cases outside endemic areas), the most plausible explanation is that the index case became infected through contact with an animal or human infected with MPXV while leaving or visiting the endemic West African countries. Then, it was seemingly coincidentally introduced into an MSM community and continued circulating over there. Another hypothesis it is that the virus was already circulating undetected for quite a while. However, this is less likely, considering the known clinical features of the disease, showing localized or generalized skin lesions. The loss of cross-protective immunity to monkeypox induced by smallpox vaccination, international travel, attendance to large gatherings, and sexual networks may explain the global spread of monkeypox infections. Although human monkeypox remains rare and the risk to the general public is low, the outbreak was declared a PHEIC because it is an extraordinary event, constitutes a public health international risk, and a coordinated international response is required.

## Data Availability

Not applicable.

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
