# Peer review of "Monkeypox Mysteries of the New Outbreak in Non-Endemic Areas"

_ijerph, 2022, doi:10.3390/ijerph192214881_

Round 1

Reviewer 1 Report (Previous Reviewer 1)

The authors have addressed my concerns. I don't have any other questions.

Author Response

Response to Reviewer #1 comments:

Comment: The authors have addressed my concerns. I don't have any other questions.

Response:

The authors thank for the relevant comments that help us to improve the manuscript

Reviewer 2 Report (New Reviewer)

Manuscript ID: ijerph-2008159

Title: Monkeypox mysteries of the new outbreak in non-endemic ar-2 eas

F. Antunes and A. Virgolino

Overview and general recommendation:

The recent spread of monkeypox around the world has caused concern to WHO and health authorities in many countries. This monkeypox outbreak has unusual features, which once again drew attention to orthopoxviruses. The authors of the manuscript have tried to collect relevant facts and explain the features of this new outbreak. So, this study is on a topic of relevance and general interest to the readers of the journal.

The manuscript is well written and there are only two minor comments:

1.      It should be pointed out the discovery of this virus in Copenhagen in 1958 (von Mangus et al., 1959).

2.      As the authors describe various clades and variants of the virus, it would be useful to provide a link to the first complete genome of monkeypox virus (Schelkunov et al., 2001; 2002).

Author Response

Response to Reviewer #2 comments:

Comment: It should be pointed out the discovery of this virus in Copenhagen in 1958 (von Mangus et al., 1959).

Response:

Thank you for the suggestion. The following sentence was added in the first paragraph “The virus was first discovered at an animal facility in Copenhagen, Denmark, in 1958 [1]”, and the following reference was added “1. von Magnus, P.; Andersen, E.K.; Petersen, K.B.; Birch-Andersen, A. A Pox-Like Disease in Cynomolgus Monkeys. Acta Pathol Microbiol Immunol Scand A. 1959;46(2):156-176.”

Comment: As the authors describe various clades and variants of the virus, it would be useful to provide a link to the first complete genome of monkeypox virus (Schelkunov et al., 2001; 2002).

Response:

Thank you for the suggestion. The following sentence was added in the first paragraph “Genome sequence of MPXV confirmed as a distinct species of orthopoxvirus that is not a direct ancestor or a direct descendent of VARV [2]”, and the following reference was added “2. Shchelkunov, S.N.; Totmenin, A.V.; Safronov, P.F.; Mikheev, M.V.; Gutorov, V.V.; Ryazankina, O.I.; Petrov, N.A.; Babkin, I.V.; Uvarova, E.A.; Sandakhchiev, L.S.; et al. Analysis of the monkeypox virus genome. Virology. 2002;297(2):172-94.”

This manuscript is a resubmission of an earlier submission. The following is a list of the peer review reports and author responses from that submission.

Round 1

Reviewer 1 Report

The manuscript by Francisco et al summarized the reported information on monkeypox from different aspects and raised some important questions which need to be solved. Since the UK case was reported in May 2022, almost 60,000 cases have been reported worldwide as of 13 Sep 2022. The number of confirmed infected cases is increasing at a rapid rate, so WHO Director-General has declared the ongoing monkeypox outbreak a public health emergency of international concern. For the sudden monkeypox outbreak outside endemic areas, it is very important to figure out how the outbreak started and why monkeypox became so transmissible among humans. This will be very helpful for blocking the transmission of monkeypox. As described in this paper, gene mutations, smallpox vaccination discontinuation and so on may contribute to the monkeypox outbreak. Overall, this is a nice paper which can give us much useful information. I only have several minor questions.

1.       Page 1, line 22, I think it is better to describe as “variola virus (VARV), the causative agent of smallpox”.

2.       Page 1, line 27, “humans and wild animals, reservoir(s) is common”, it is better to describe as “humans and wild animals or reservoir(s) is common”.

3.       Page 1, line 32-34, the time is wrong. Should be “In May 2022, the multi-country outbreak of human monkeypox which started to spread to Europe and North America, and in July 2022 was determined a Public Health Emergency of International Concern (PHEIC) by the World Health Organization (WHO)”.

4.       Page 2, line 48-49, this description is not right. Some deaths have been reported. Please see this report https://journals.sagepub.com/doi/10.1177/20499361221124027.  Should be “This current global outbreak, with thousands of cases, till now, fortunately, has led to only a few deaths or serious cases which require hospital admission”.

5.       Page 2, line 68-70, please show where this information is from, I think from the WHO website https://www.who.int/news/item/12-08-2022-monkeypox--experts-give-virus-variants-new-names, please cite.

6.       Page 3, line 114-116, this study has been done in this paper https://doi.org/10.1016/S1473-3099(22)00513-8, please update the manuscript.

Author Response

Comment: The manuscript by Francisco et al summarized the reported information on monkeypox from different aspects and raised some important questions which need to be solved. Since the UK case was reported in May 2022, almost 60,000 cases have been reported worldwide as of 13 Sep 2022. The number of confirmed infected cases is increasing at a rapid rate, so WHO Director-General has declared the ongoing monkeypox outbreak a public health emergency of international concern. For the sudden monkeypox outbreak outside endemic areas, it is very important to figure out how the outbreak started and why monkeypox became so transmissible among humans. This will be very helpful for blocking the transmission of monkeypox. As described in this paper, gene mutations, smallpox vaccination discontinuation and so on may contribute to the monkeypox outbreak. Overall, this is a nice paper which can give us much useful information. I only have several minor questions.

Response:

The authors thank for the reviewer’s encouraging comments.

Comment: 1. Page 1, line 22, I think it is better to describe as “variola virus (VARV), the causative agent of smallpox”.

Response:

Following the reviewer’s suggestion, the sentence was changed.

Comment: 2. Page 1, line 27, “humans and wild animals, reservoir(s) is common”, it is better to describe as “humans and wild animals or reservoir(s) is common”.

Response:

As suggested, we made the change in the manuscript.

Comment: 3. Page 1, line 32-34, the time is wrong. Should be “In May 2022, the multi-country outbreak of human monkeypox which started to spread to Europe and North America, and in July 2022 was determined a Public Health Emergency of International Concern (PHEIC) by the World Health Organization (WHO)”.

Response:

We thank for the relevant comment. The correction was made in the text.

Comment: 4. Page 2, line 48-49, this description is not right. Some deaths have been reported. Please see this report https://journals.sagepub.com/doi/10.1177/20499361221124027.  Should be “This current global outbreak, with thousands of cases, till now, fortunately, has led to only a few deaths or serious cases which require hospital admission”.

Response:

The text was changed according to the suggestion.

Comment: 5. Page 2, line 68-70, please show where this information is from, I think from the WHO website https://www.who.int/news/item/12-08-2022-monkeypox--experts-give-virus-variants-new-names, please cite.

Response:

The reviewer is correct. The sentences were revised and the reference was added.

Comment: 6. Page 3, line 114-116, this study has been done in this paper https://doi.org/10.1016/S1473-3099(22)00513-8, please update the manuscript.

Response:

This part of the text was revised and the indicated reference was added to the manuscript.

Reviewer 2 Report

Based on the abstract, I understand that the aim of the current review was to clarify some major research gaps in monkeypox knowledge to strengthen the outbreak response. 

The review summarizes some aspects of the current outbreak (such as its larger size, and mutations in the virus clade) and contrasts those aspects with  outbreaks in endemic regions.

In my opinion, the paper in its current form represents a minimal contribution to the field of monkeypox research because of both its content as well as the form:

1. The authors should state the aim of the review in the introduction

2. The manuscript is not well-organized:

2.a) avoid repetitions (line 39-40/51-51/45/60-62; line 73/45-46; line 66-69/81)

2.b) text within paragraphs should be organized logically: line 89-90 starts with a sentence about MSM, then switches to smallpox vaccination, until 110 where it continues about MSM.

2.c) some statements are not well founded or incorrect (line 82: insinuate that mutations in the virus made it more transmissible - which has not yet been confirmed; line 94-97: reference 21 does not state that smallpox vaccination provides protection for 3-5 years; line 118 implies that the current epidemic is caused and sustained by mass gatherings, which may not be true; conclusion lines 135-138: the authors propose that one scenario is more plausible than the other without explaining why they think so;)

3. The authors acknowledge on several occasions that WHO gave new names to the monkeypox clades. They should use these new names throughout the mansucript.

Author Response

Comment: Based on the abstract, I understand that the aim of the current review was to clarify some major research gaps in monkeypox knowledge to strengthen the outbreak response.

The review summarizes some aspects of the current outbreak (such as its larger size, and mutations in the virus clade) and contrasts those aspects with outbreaks in endemic regions.

In my opinion, the paper in its current form represents a minimal contribution to the field of monkeypox research because of both its content as well as the form:

Response:

We thank the reviewer for the valuable inputs made to the mmanuscript. The document was thouroughly revised and we believe that all the major shortcomings were addressed.

Comment: 1. The authors should state the aim of the review in the introduction

Response:

Thank you again for the suggestion. The following sentence was added in the end of the first paragraph with the main aim of this review: “Based on available data, the aim of this review is to help understand current major gaps in the dynamics of viral transmission and epidemiological characteristics of this 2022 human monkeypox outbreak”.

Comment: 2. The manuscript is not well-organized:

Response:

We understand the concern and we tried to address, point by point, each of the comments made.

Comment: 2.a) avoid repetitions (line 39-40/51-51/45/60-62; line 73/45-46; line 66-69/81)

Response:

The following was done:

- Lines 51-53 – deleted.

- Lines 60-62 – The sentence was changed to “Characterization of all MPXV ongoing outbreak genome sequences, confirming a linkage between the 2022 MPXV lineage with the clade IIa (former Western African clade), suggests that the outbreak had a single origin [13]”.

- Lines 73-75 – The sentence was changed to “Second, as mentioned, the number of monkeypox cases detected in this 2022 human monkeypox outbreak, surpassed those in monkeypox-endemic regions of West and Central Africa [11]”.

- Lines 81-82 – The sentence was changed to “The new variant, clade IIb, with several mutations than would be expected, may promote changes in the biologic aspects of the virus, including increased transmissibility, and together with super spreader events, and travel abroad likely triggered the rapid worldwide dissemination [12,13]”.

Comment: 2.b) text within paragraphs should be organized logically: line 89-90 starts with a sentence about MSM, then switches to smallpox vaccination, until 110 where it continues about MSM.

Response:

- Lines 89 and 90 – The sentence was changed to “Third, almost all the human monkeypox case clusters included men aged 20-50 years [12,19,20]”.

Comment: 2.c) some statements are not well founded or incorrect (line 82: insinuate that mutations in the virus made it more transmissible - which has not yet been confirmed; line 94-97: reference 21 does not state that smallpox vaccination provides protection for 3-5 years; line 118 implies that the current epidemic is caused and sustained by mass gatherings, which may not be true; conclusion lines 135-138: the authors propose that one scenario is more plausible than the other without explaining why they think so;)

Response:

- Lines 93-97 – The sentence was changed to “The end of the global smallpox vaccination campaign in 1977 has resulted in a susceptible population because most estimates suggest immunity from the smallpox vaccination lasts three to five years, and decreasing immunity thereafter, and beyond that time its ability to protect against human monkeypox also may decrease, and the human-to-human cycles of transmission can occur”

- The references Cohen J et al. Science 2001;294(5544):985 and World Health Organization. WHO Expert Committee on Smallpox Eradication. World Health Organization Technical Report Series nº 493, Geneva 1972 – were added.

- Lines 118-121 – The sentence was changed to “The explanation for this pattern of the 2022 human monkeypox outbreak, could be the attendance at large gatherings linked to sex-onsite activities and international travel, probably starting when the virus was introduced in a MSM community, thus spreading easily in a non-immune population, and amplified through sexual networks [25].”

- Lines 134-138 – The sentence was changed to “How the ongoing outbreak started remains unclear, however, giving the phylogenomic characterization of the virus and the MPVX historical epidemiology (rarity of cases outside endemic areas), the most plausible explanation is that the index case became infected through contact with an animal or human infected with MPXV while leaving or visiting the endemic West African countries. Then coincidentally introduced into an MSM community and continued circulating over there. Another hypothesis, it is that the virus was already circulating undetected for quite a while. However, this is less likely, considering the known clinical features of the disease, showing localized or generalized skin lesions.”

Comment: 3. The authors acknowledge on several occasions that WHO gave new names to the monkeypox clades. They should use these new names throughout the mansucript.

Response:

The new names of the MPXV were used throughout the manuscript, as suggested.